DATA RELEASE

# Bicolor angelfish (*Centropyge bicolor*) provides the first chromosome-level genome of the Pomacanthidae family

Chunhua Li[1,†], Xianwei Yang[1,3,†], Libin Shao[1,†], Rui Zhang[1], Qun Liu[1], Mengqi Zhang[1], Shanshan Liu[1], Shanshan Pan[1], Weizhen Xue[1], Congyan Wang[1], Chunyan Mao[1], He Zhang[1,2,*] and Guangyi Fan[1,*]

1 BGI-Qingdao, BGI-Shenzhen, Qingdao 26655-5, China
2 Department of Biology, Hong Kong Baptist University, Hong Kong, China
3 College of Life Sciences, University of Chinese Academy of Sciences, Beijing 100049, China

## ABSTRACT

The Bicolor Angelfish, *Centropyge bicolor*, is a tropical coral reef fish. It is named for its striking two-color body. However, a lack of high-quality genomic data means little is known about the genome of this species. Here, we present a chromosome-level *C. bicolor* genome constructed using Hi-C data. The assembled genome is 650 Mbp in size, with a scaffold N50 value of 4.4 Mbp, and a contig N50 value of 114 Kbp. Protein-coding genes numbering 21,774 were annotated. Our analysis will help others to choose the most appropriate *de novo* genome sequencing strategy based on resources and target applications. To the best of our knowledge, this is the first chromosome-level genome for the Pomacanthidae family, which might contribute to further studies exploring coral reef fish evolution, diversity and conservation.

**Subjects** Genetics and Genomics, Evolutionary Biology, Marine Biology

**Submitted:**   14 May 2021

\* Corresponding authors. E-mail: fanguangyi@genomics.cn; zhanghe@genomics.cn

† Contributed equally.

Preprint submitted at https://doi.org/10.1101/2021.10.24.465606

# DATA DESCRIPTION

## Background

*Centropyge bicolor* (NCBI:txid109723; FishbaseID: 5454; urn:lsid:marinespecies.org:taxname:211780) (Figure 1), also known as the Bicolor, Two-Colored, or Pacific Rock Beauty Angelfish, is a showy coral reef fish commonly distributed in the Indo–Pacific ocean (from East Africa to the Samoan and Phoenix Islands, north to southern Japan, south to New Caledonia; throughout Micronesia). As a member of the Pomacanthidae family, it is similar to those of the Chaetodontidae (Butterflyfishes) but is distinguished by the presence of strong preopercle spines. *C. bicolor* has clear boundaries between its body colors, so might be a good model in which to study body color development in coral fish [1].

## Context

Although the availability of genetic, and especially genomic resources, remains limited for the Pomacanthidae family, we assembled the first *C. bicolor* reference genome. This will provide valuable information for genetic studies of this coral reef fish, and will contribute to studies in body color diversity. With the whole genome sequence of *C. bicolor*, it might be

**Figure 1.** Photograph of *Centropyge bicolor.*

**Figure 2.** Protocols for BGISEQ-500, stLFR and Hi-C library preparation and construction, and genome assembly, for the Bicolor Angelfish, *Centropyge bicolor* [2]. https://www.protocols.io/widgets/doi?uri=dx.doi.org/10.17504/protocols.io.bpxhmpj6

possible to explore the genetic mechanisms of body color development in coral reef fish by comparative genomic methods.

## METHODS AND RESULTS

A protocols collection for BGISEQ-500, stLFR and Hi-C library construction is available in protocols.io (Figure 2) [2].

## Sample collection and genome sequencing

A *C. bicolor* individual was collected from the market in Xiamen, Fujian Province, China. DNA was extracted from fresh muscle tissue according to a standard protocol. Single-tube long fragment read (stLFR) [2] and Hi-C libraries were constructed following the

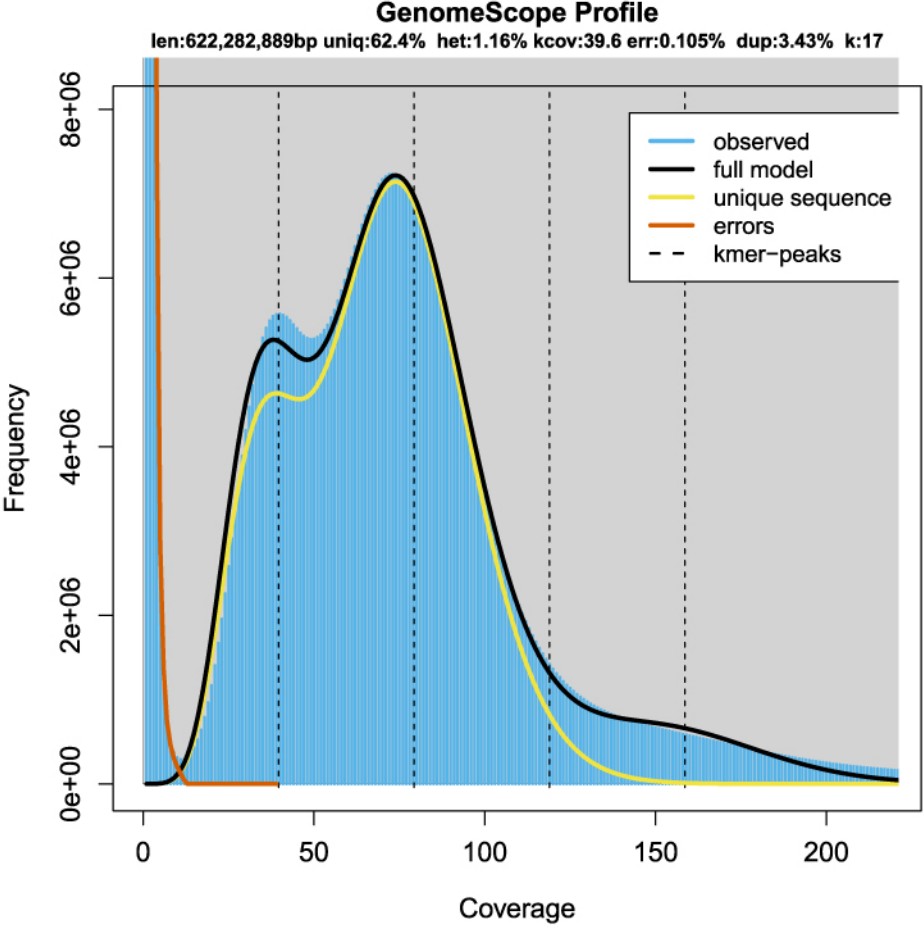

**Figure 3.** The 17-mer depth distribution of *Centropyge bicolor*. The estimated genome size is 662.27 Mbp and the heterozygosity is 1.16%.

manufacturers' instructions [2, 3] to sequence and assemble the genome. We obtained 130.47 Gbp (gigabase pairs; ~197×) raw stLFR data and 134.57 Gbp (~203.20×) raw Hi-C data (Table 1) using the BGISEQ-500 platform in 100-bp (basepair) paired-end mode.

Low-quality reads (sequences with more than 40% of bases with a quality score lower than 8), polymerase chain reaction (PCR) duplications, adaptor sequences and reads with a high (greater than 10%) proportion of ambiguous bases (Ns) occurring in stLFR data were filtered using SOAPnuke (v1.6.5; RRID:SCR_015025) [4]. We obtained 62.6 Gbp (~91.67×) clean data (Table 1) to assemble the draft genome. Meanwhile, HiC-Pro (v. 2.8.0) [5] was used for the quality control of raw Hi-C data, and 42.51 Gbp (~64.19×) valid data were used to assemble the genome to the chromosome-level (Table 1).

## Genome assembly

Using GenomeScope software (RRID:SCR_017014) with stLFR clean data, *k*-mer distribution was used to understand the genome complexity before genome assembly [6]. The genome size of *C. bicolor* was estimated as 662.27 Mbp (megabase pairs), with 37.6% repeat sequences and 1.16% heterozygous sites (Table 2, Figure 3).

**Table 1.** Statistics of DNA sequencing data.

| Libraries | Read length | Raw data | | Valid data | |
|---|---|---|---|---|---|
| | | Total bases (Gbp) | Sequencing depth (×) | Total bases (Gbp) | Sequencing depth (×) |
| stLFR | 100:100 | 130.47 | 197.00 | 60.71 | 91.67 |
| Hi-C | 100:100 | 134.57 | 203.20 | 42.51 | 64.19 |

Sequencing depth = Total bases / Genome size, where the genome size is the result of *k*-mer estimation, as shown in Table 2.

**Table 2.** Statistical information of 17-mer analysis.

| *k*-mer | *k*-mer number | *k*-mer Depth | Heterozygosity (%) | Genome size (Mbp) |
|---|---|---|---|---|
| 17 | 50,994,645,240 | 77 | 1.16 | 662.27 |

The genome size, G, was defined as $G = K_{num}/K_{depth}$, where $K_{num}$ is the total number of *k*-mers, and $K_{depth}$ is the most frequently occurring *k*-mer.

**Table 3.** Statistics of the draft assembly with stLFR data.

| Statistics | Contig | Scaffold |
|---|---|---|
| Total number (#) | 40,442 | 29,065 |
| Total length (bp) | 655,705,062 | 681,285,455 |
| Gap (N) (bp) | 0 | 25,580,393 |
| Average length (bp) | 16,213.47 | 23,440.06 |
| N50 length (bp) | 115,524 | 4,424,004 |
| N90 length (bp) | 6,029 | 7,618 |
| Maximum length (bp) | 1,148,507 | 21,943,074 |
| Minimum length (bp) | 48 | 940 |
| GC content (%) | 41.74 | 41.74 |

We reformatted the clean stLFR data into 10× Genomics format using an in-house script [7] and assembled the draft genome using Supernova (v.2.0.1, RRID:SCR_016756) [8] with default parameters. The draft genome was 681 Mbp, with a contig N50 of 115.5 Kbp (kilobase pairs) and scaffold N50 of 4.4 Mbp (Table 3), which is similar to the estimated genome size.

To obtain the chromosome-level genome, we used Juicer (v3, RRID:SCR_017226) [9] to build a contact matrix and 3dDNA (v.170123) [10] to sort and anchor scaffolds with the parameters: "–m haploid –s 4 –c 24". There are 24 distinct contact blocks, which correspond to 24 chromosomes, representing 96% of the whole genome (Figures 4A, 5, Table 4). On evaluating the completeness of the genome and gene set using Benchmarking Universal Single-Copy Orthologs (BUSCO, v.3.0.2, RRID:SCR_015008) [11] and a vertebrata database, our assembly maintained a score of 96.2% (Table 5). We also identified putative homologous chromosomal regions between *C. bicolor* and *Oryzias latipes* by MCscanx [12] (Figure 6).

In addition, we cut off partial stLFR reads (25 M) for assembly by MitoZ with default parameters [13] and obtained a 16,961-bp circular mitochondrial genome of *C. bicolor*. Thirteen protein-coding genes, 24 tRNA genes and three rRNA genes were annotated by GeSeq (RRID:SCR_017336) [14] (Figure 4B).

## Genomic annotation

For the annotation of repeats, we carried out homolog annotation and *ab initio* prediction independently. RepeatMasker (v.4.0.6, RRID:SCR_012954) [15], RepeatProteinMask (a module from RepeatMasker) and trf (Tandem Repeats Finder, v.4.07b) [16] were used to

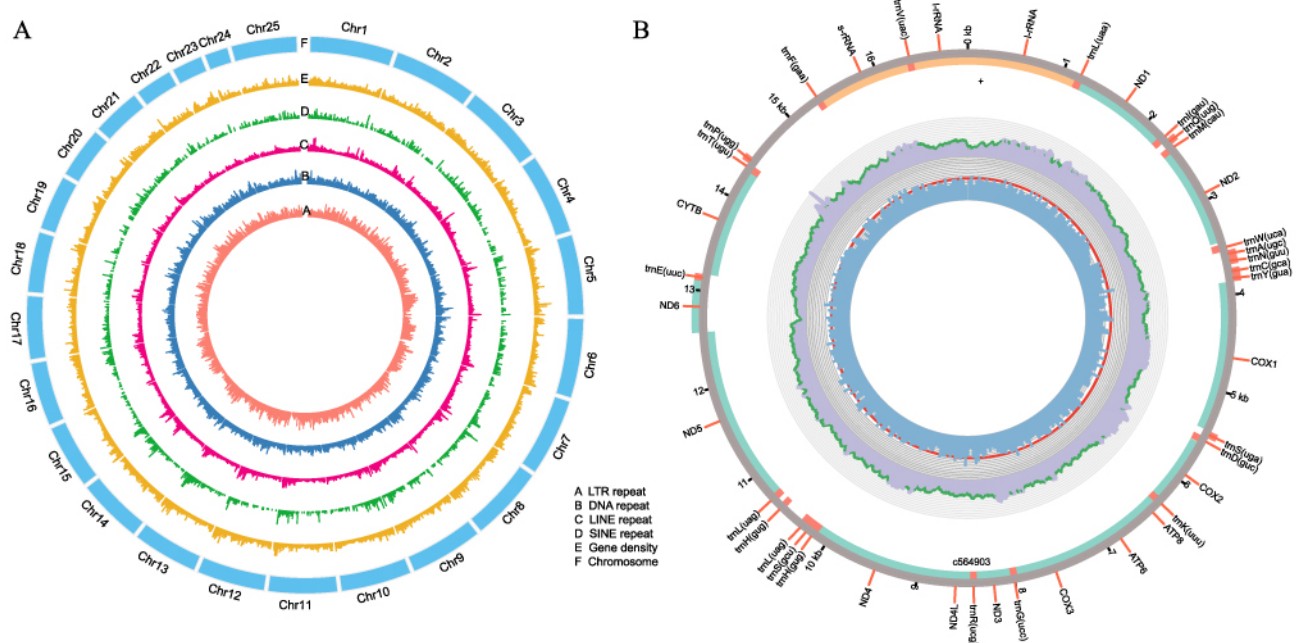

**Figure 4.** Annotation of the *Centropyge bicolor* genome. (A) Basic genomic elements of the *Centropyge bicolor* genome. LTR, long terminal repeat; LINE, long interspersed nuclear elements; SINE, short interspersed elements. (B) Physical map of mitochondrial assembly.

**Table 4.** Statistics of the chromosome-level genome.

| Statistics | Contig | Scaffold |
|---|---|---|
| Total number (#) | 40,778 | 28,555 |
| Total length (bp) | 655,705,062 | 680,873,932 |
| Gap (N) (bp) | 0 | 25,168,870 |
| Average length (bp) | 16,079.87 | 23,844.30 |
| N50 length (bp) | 113,563 | 21,943,074 |
| N90 length (bp) | 5,988 | 7,542 |
| Maximum length (bp) | 1,148,507 | 28,105,280 |
| Minimum length (bp) | 43 | 43 |
| GC content (%) | 41.74 | 41.74 |

identify known repetitive sequences by comparing the whole genome with RepBase [17]. LTR_FINDER (v.1.06, RRID:SCR_015247) [16, 18] and RepeatModeler (v.1.0.8, RRID:SCR_015027) [19] were used in *de novo* prediction. We also classified transposable elements (TEs) from the integration of all repeats. In total, we identified 124 Mbp (18.32% of the entire genome) of repetitive sequences (Figure 4A, Table 6), including 110 Mbp of TEs (Figure 4A, Table 7).

Homolog-based and *ab initio* prediction were used to identify the protein-coding genes. Augustus (v.3.3, RRID:SCR_008417) [20] was used in *ab initio* prediction basing on a repeat-masked genome [21]. Protein sequences of *Astatotilapia calliptera*, *Danio rerio*, *Larimichthys crocea*, and *Oreochromis niloticus* were downloaded from the National Center for Biotechnology Information (NCBI) GenBank database and aligned to the *C. bicolor* genome for homolog gene annotation with Genewise (v2.4.1, RRID:SCR_015054) [22]. Finally, we used GLEAN [23] to integrate all the above evidence and obtained a total of 21,774 genes,

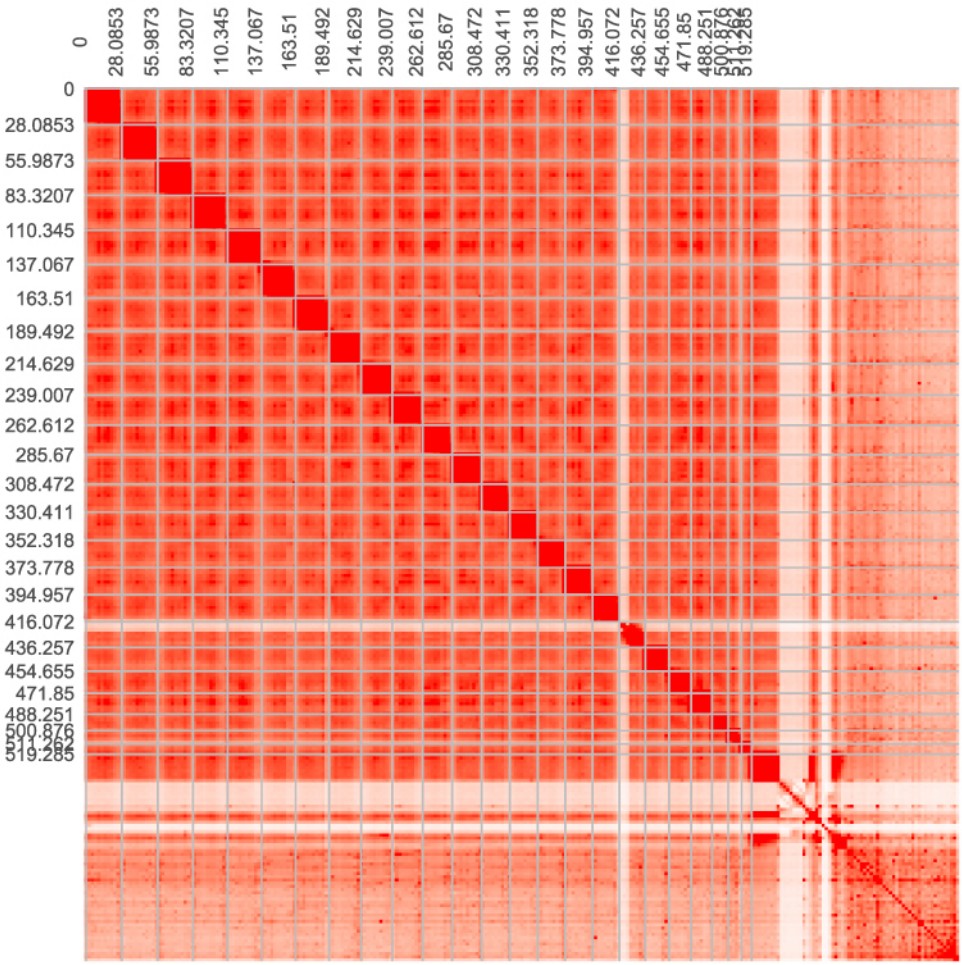

**Figure 5.** Heat map of interactive intensity between chromosome sequences.

**Table 5.** Statistics of the BUSCO assessment.

| Types of BUSCOs | Gene set | | Assembly | |
|---|---|---|---|---|
| | Number | Percentage (%) | Number | Percentage (%) |
| Complete BUSCOs | 2,408 | 93.1 | 2,486 | 96.2 |
| Complete single-copy BUSCOs | 2,348 | 90.8 | 2,438 | 94.3 |
| Fragmented BUSCOs | 81 | 3.1 | 64 | 2.5 |
| Missing BUSCOs | 97 | 3.8 | 36 | 1.3 |
| Total BUSCO groups searched | 2,586 | 100 | 2,586 | 100 |

**Table 6.** Statistics of repetitive sequences.

| Type | Repeat size (bp) | Percentage of genome (%) |
|---|---|---|
| TRF | 14,165,095 | 2.08 |
| RepeatMasker | 43,423,877 | 6.38 |
| RepeatProteinMask | 12,503,750 | 1.84 |
| *De novo* | 110,871,693 | 16.28 |
| Total | 124,708,977 | 18.32 |

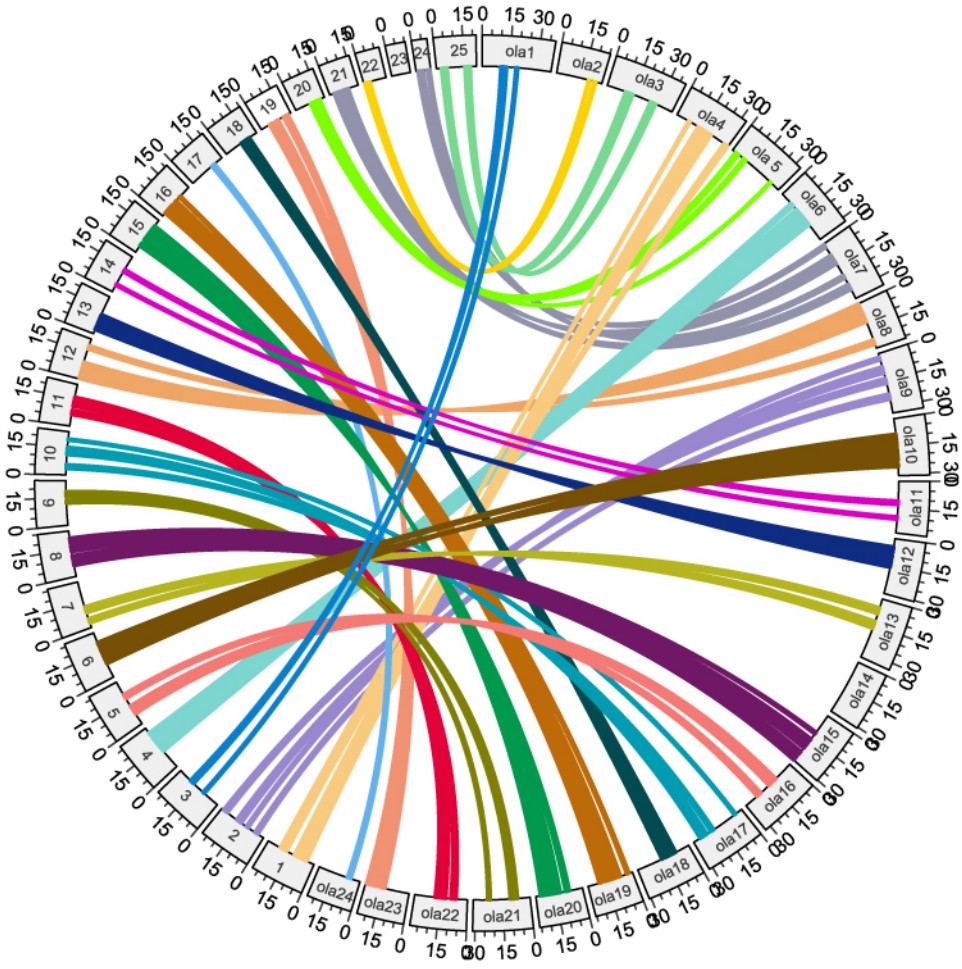

**Figure 6.** Homologous chromosomal regions between *Centropyge bicolor* and *Oryzias latipes*.

**Table 7.** Statistics of transposable elements.

|  | Repbase TEs, *n* (%) | Protein TEs, *n* (%) | *De novo* TEs, *n* (%) | Combined TEs, *n* (%) |
|---|---|---|---|---|
| DNA | 27,163,851 (3.990) | 1,068,990 (0.157) | 61,731,447 (9.067) | 70,925,963 (10.417) |
| LINE | 10,228,332 (1.502) | 6,956,340 (1.022) | 20,006,579 (2.938) | 26,714,285 (3.924) |
| SINE | 856,125 (0.126) | 0 (0.000) | 497,024 (0.073) | 1,187,676 (0.174) |
| LTR | 10,971,817 (1.611) | 4,485,808 (0.659) | 16,270,071 (2.390) | 23,101,529 (3.393) |
| Other | 10,041 (0.001) | 0 | 0 | 10,041 (0.001) |
| Unknown | 0 | 0 | 14,054,230 (2.064) | 14,054,230 (2.064) |
| Total | 43,423,877 (6.378) | 12,503,750 (1.836) | 99,265,690 (14.579) | 109,868,166 (16.136) |

which contained 11 exons on average and had an average coding sequence (CDS) length of 1,575 bp (Table 8).

To predict gene functions, 21,774 genes were aligned against several public databases, including TrEMBL [24], SwissProt [24], KEGGViewer [25] and InterProScan [26]. As a result, 99.67% of all genes were predicted functionally (Table 9, Figure 7).

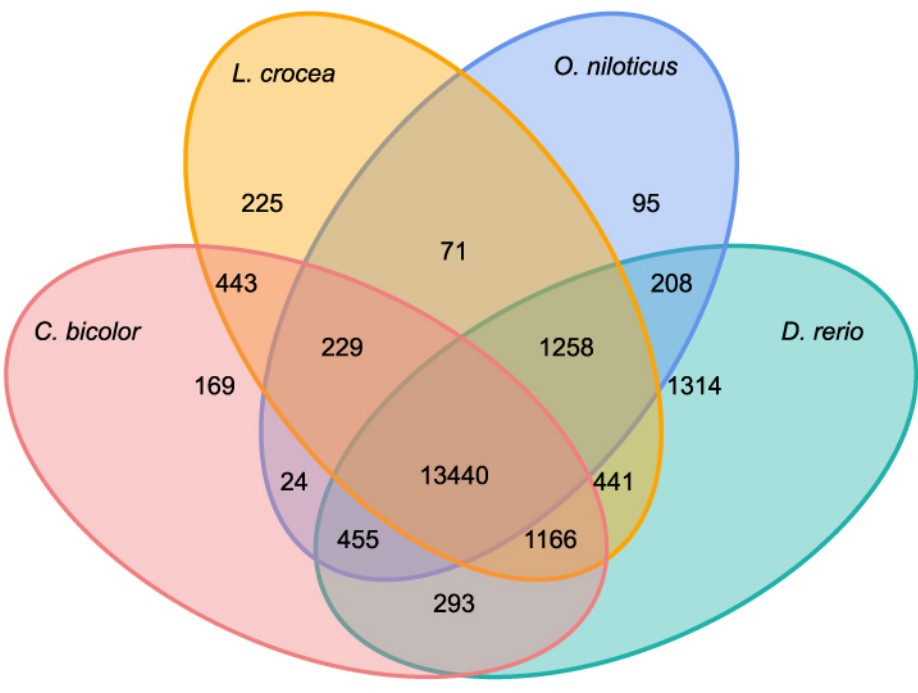

**Figure 7.** Venn diagram of orthologous gene families. Four teleost species (*Centropyge. bicolor, Larimichthys crocea, Oreochromis niloticus, and Danio rerio*) were used to generate the Venn diagram based on gene family cluster analysis.

**Table 8.** Statistics of the predicted genes in the bicolor angelfish genome.

|  | Gene set | Gene number | Average transcript length (bp) | Average CDS length (bp) | Average intron length (bp) | Average exon length (bp) | Average exons per gene |
|---|---|---|---|---|---|---|---|
| Homolog | *Astatotilapia calliptera* | 51,174 | 21,762.29 | 2,259.23 | 1,691.33 | 180.29 | 12.53 |
|  | *Danio rerio* | 22,005 | 27,982.75 | 1,570.36 | 3,438.82 | 180.90 | 8.68 |
|  | *Larimichthys crocea* | 47,419 | 19,884.78 | 2,139.39 | 1,575.94 | 174.50 | 12.26 |
|  | *Oreochromis niloticus* | 47,067 | 17,771.04 | 1,906.97 | 1,608.29 | 175.53 | 10.86 |
| *De novo* | Augustus | 34,470 | 9,675.42 | 1,335.20 | 1,344.81 | 185.40 | 7.20 |
| GLEAN |  | 21,774 | 14,024.40 | 1,906.28 | 1,206.07 | 172.55 | 11.05 |

The GLEAN gene set is the integrated result of *de novo* gene predictions and homolog gene predictions.

## Phylogenetic analysis

We downloaded the gene data of seven representative teleost fishes from NCBI to study the phylogenetic relationships between *C. bicolor*. These seven fishes were: *Danio rerio, Gasterosteus aculeatus, Gadus morhua, Larimichthys crocea, Oryzias latipes, Oreochromis niloticus* and *Tetraodon nigroviridis*. For each dataset, the longest transcripts were selected and aligned to each other by BLASTP (v2.9.0, RRID:SCR_001010) [27] (*E*-value ≤ 1e-5). TreeFam (v.2.0.9, RRID:SCR _013401) [28] was used to cluster gene families, with default parameters. Among all 20,706 clustered gene families, there were 4,450 common single-copy families and 57 families specific to *C. bicolor* (Table 10). With single-copy sequences, we used PhyML (v.3.3, RRID:SCR_014629) [29] to construct the phylogenetic tree of *C. bicolor* and the seven other fishes mentioned above, setting *D. rerio* as an outgroup.

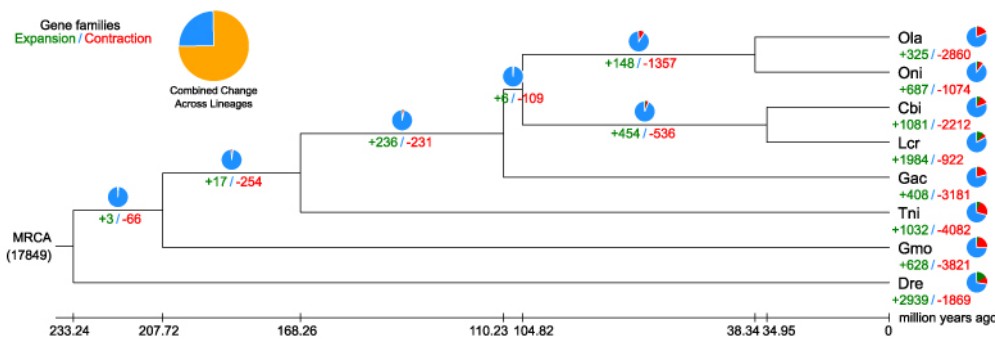

**Figure 8.** Comparative analysis of the *Centropyge bicolor* genome. (A) The protein-coding genes of the eight species were clustered into 17,849 gene families. Among these gene families, 4,450 were single-copy gene families. (B) Phylogenetic analysis of *Centropyge bicolor* (Cbi.), *Danio rerio* (Dre.), *Gasterosteus aculeatus* (Gac.), *Gadus morhua* (Gmo.), *Larimichthys crocea* (Lcr.), *Oryzias latipes* (Ola.), *Oreochromis niloticus* (Oni.), and *Tetraodon nigroviridis* (Tni.) using single-copy gene families. The species differentiation time between *Centropyge bicolor* and *Larimichthys crocea* was ~34.95 million years.

**Table 9.** Statistics of the functional annotation.

| Database | Number | Percentage (%) |
|----------|--------|----------------|
| Total | 21,774 | 100.00 |
| SwissProt | 20,784 | 95.45 |
| KEGG | 19,168 | 88.03 |
| TrEMBL | 21,688 | 99.61 |
| Interpro | 20,153 | 92.56 |
| Overall | 21,702 | 99.67 |

**Table 10.** Statistics of gene family clustering.

| Species | Total genes | Unclustered genes | Families | Unique families | Average number of genes per family |
|---------|-------------|-------------------|----------|-----------------|-------------------------------------|
| *Centropyge bicolor* | 21,774 | 694 | 16,219 | 57 | 1.3 |
| *Danio rerio* | 30,067 | 2,188 | 18,575 | 726 | 1.5 |
| *Gasterosteus aculeatus* | 20,756 | 784 | 15,921 | 16 | 1.25 |
| *Gadus morhua* | 19,987 | 535 | 15,630 | 9 | 1.24 |
| *Larimichthys crocea* | 24,403 | 610 | 17,273 | 55 | 1.38 |
| *Oryzias latipes* | 19,535 | 1,048 | 14,805 | 87 | 1.25 |
| *Oreochromis niloticus* | 21,431 | 180 | 15,780 | 14 | 1.35 |
| *Tetraodon nigroviridis* | 19,544 | 901 | 14,803 | 57 | 1.26 |

Based on the phylogenetic tree and single-copy sequences, the divergence time between different species was estimated by MCMCTREE with parameters of "–model 0 –rootage 500 -clock 3". The results showed that *C. bicolor* was formed ~34.95 million years ago, when differentiated from the common ancestor with *L. crocea* (Figure 8).

## Analysis of bicolor formation in teleosts

Current studies suggest that different pigment cells produce different pigments. Some types of pigment cells already have been identified in teleost [30]. *C. bicolor* has an attractive body color with clear color boundaries, but the molecular mechanism underlying this remains unknown. Compared with other teleost, there are 1,081 expanded gene families and 57 specific gene families in *C. bicolor* (Figure 9). Functional enrichment analysis

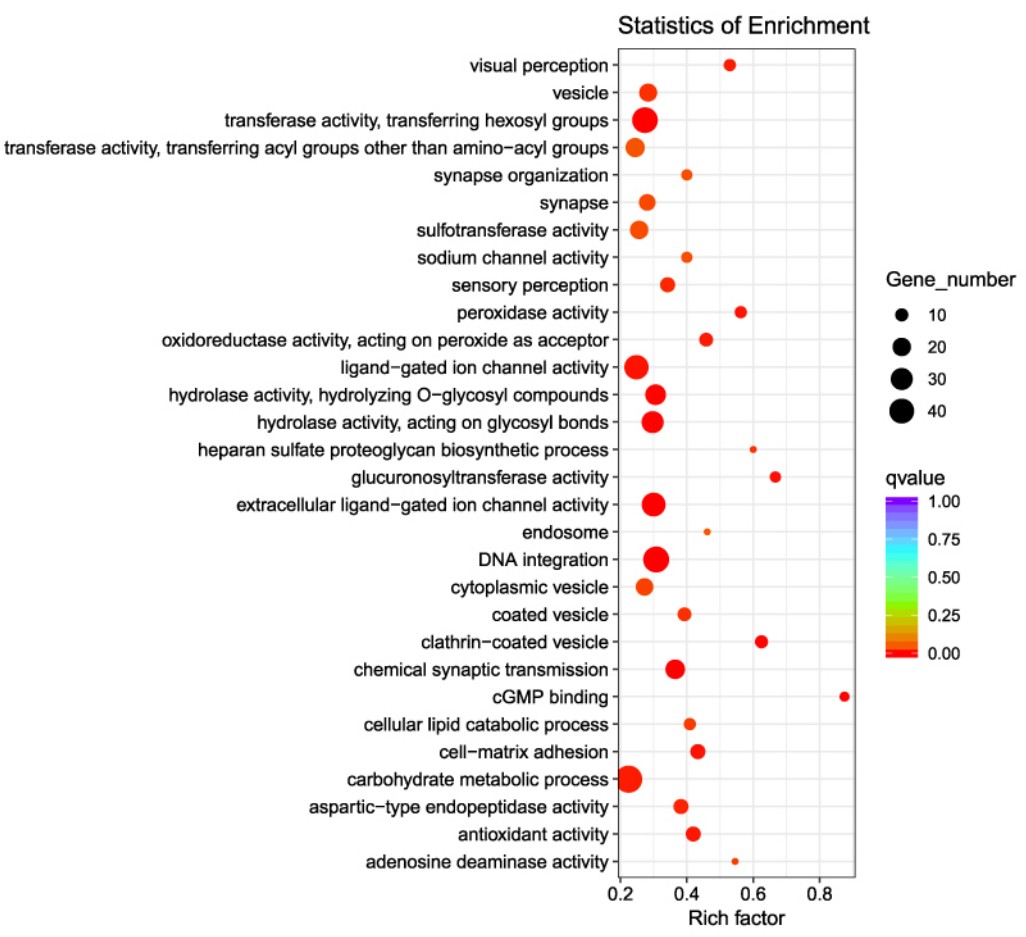

**Figure 9.** Statistics of gene function enrichment (Gene Ontology) for expanded genes of *Centropyge bicolor*. Nodes are colored by *q*-value (adjusted *p*-value). Node size is shown according to its enriched gene number.

showed that notable expansion occurred in those gene families related to visual development and enzyme metabolism (Figure 9).

## RE-USE POTENTIAL

Coral reef fishes, with distinctive color patterns and color morphs, are important for understanding the adaptive evolution of fishes. In this study, we firstly assembled a high-quality, chromosome-level genome of *C. bicolor*, with a length of 681 Mbp, and annotated 21,774 genes. This is the first genome of a fish from the Pomacanthidae family. These genomic data will be useful for genome-scale comparisons and further studies on the mechanisms underlying colorful body development and adaptation.

## DATA AVAILABILITY

The data sets supporting the results of this article are available in the *GigaScience* Database [31]. Raw reads from genome sequencing and assembly are deposited at the China National Gene Bank under reference number CNP0001160, which contains sample

information (CNS0315939), Hi-C raw data (CNX0286336) and stLFR raw data (CNX0286337). The project also has been deposited at NCBI under accession ID PRJNA702283.

## DECLARATIONS
## LIST OF ABBREVIATIONS

bp: base pair; BUSCO: Benchmarking Universal Single-Copy Orthologs; Gbp: gigabase pair; Kbp: kilobase pair; KEGG: Kyoto Enyclopedia of Genes and Genomes; Mbp: megabase pair; NCBI: National Center for Biotechnology Information; stLFR: single-tube long fragment reads; TE: transposable element.

## ETHICAL APPROVAL

All resources used in this study were approved by the Institutional Review Board of BGI (IRB approval No. FT17007). This experiment has passed the ethics audit of the Beijing Genomics Institute (BGI) Gene Bioethics and Biosecurity Review Committee.

## CONSENT FOR PUBLICATION

Not applicable.

## COMPETING INTERESTS

The authors declare that they have no competing interests.

## FUNDING

This work was supported by funding from the "Blue Granary" project for scientific and technological innovation of China (2018YFD0900301-05).

## AUTHORS' CONTRIBUTIONS

H.Z. and G.F. designed this project. M.Z. prepared the samples. S.L., S.P., W.X., C.W. and C.M. conducted the experiments. C.L., X.Y., L.S., R.Z. and Q.L. did the analyses. C.L., X.Y., L.S., R.Z. wrote and revised the manuscript. All authors read and approved the final version of the manuscript.

## ACKNOWLEDGEMENTS

We thank the China National Genebank for technical support in constructing and sequencing the stLFR library.

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
