## [Reviewer Report]

Comments on revised manuscriptI am happy with the changes made and thank the authors for addressing them. The manuscript is in my opinion acceptable for publication. Congratulations to the authors for providing a reference genome for this exciting fish species to the community.

---

## [Reviewer Report]

Reviewer name and names of any other individual's who aided in reviewer Ole K. TørresenDo you understand and agree to our policy of having open and named reviews, and having your review included with the published papers. (If no, please inform the editor that you cannot review this manuscript.)YesIs the language of sufficient quality?NoPlease add additional comments on language quality to clarify if needed
Almost every second sentence in the abstract would need work, and so is the rest of the manuscript.Are all data available and do they match the descriptions in the paper? YesAdditional CommentsAre the data and metadata consistent with relevant minimum information or reporting standards? See GigaDB checklists for examples <a href="http://gigadb.org/site/guide" target="_blank">http://gigadb.org/site/guide</a>YesAdditional CommentsIs the data acquisition clear, complete and methodologically sound?YesAdditional CommentsIs there sufficient detail in the methods and data-processing steps to allow reproduction?YesAdditional CommentsIs there sufficient data validation and statistical analyses of data quality? YesAdditional CommentsIs the validation suitable for this type of data?YesAdditional CommentsIs there sufficient information for others to reuse this dataset or integrate it with other data?YesAdditional Comments
General comments:
The authors have created a chromosome-level genome assembly of bicolor angelfish using stLFR and HiC libraries.


The language in this manuscript needs some work. After commenting on every second sentence in the abstract regarding some language matter, I saw that I couldn’t continue commenting all these matters. Please do a good clean-up in the language, so that it is easier to read. I’ll point out some issues during the manuscript, but will not find all and I can’t manage to point out all I do find.



Specific comments:
Line 19: «...special and beautiful two-color body” is a bit subjective. Maybe something like “…remarkable and striking two-color body” instead?

Line 20: I know this is the abstract, but I don’t understand what “the mechanism of bicolor body” could mean. Maybe rephrase?

Line 22: I’ve seen this many places, but it should be a lower-case k in kb, not upper case like Kb. The k stands for kilo which is a metric prefix meaning thousand (https://en.wikipedia.org/wiki/Metric_prefix).

Line 25: “As we are known,” should be “as far as we know”.

Line 27: “Future research” instead of “future researches”. 

Line 46: Which protocol are you talking about? 

Table 1: How can you end up with more “valid data” than “raw data”? Did you mix up something here? It looks consistent with the text, but there’s likely something wrong.


Any Additional Overall Comments to the AuthorRecommendationMinor Revision

---

## [Reviewer Report]

Reviewer name and names of any other individual's who aided in reviewer Claudius KratochwilDo you understand and agree to our policy of having open and named reviews, and having your review included with the published papers. (If no, please inform the editor that you cannot review this manuscript.)YesIs the language of sufficient quality?NoPlease add additional comments on language quality to clarify if needed
The text is understandable, but has many grammatical errors. The manuscript would greatly improve through language editing.Are all data available and do they match the descriptions in the paper? YesAdditional CommentsI did not check every single file, but all data I looked for I found to be publicly available. It would help if the "Availability of supporting data and materials" statement would be a bit more comprehensive. Data A is deposited under X, Data B is deposited under Y-Z instead of just providing the project ID.Are the data and metadata consistent with relevant minimum information or reporting standards? See GigaDB checklists for examples <a href="http://gigadb.org/site/guide" target="_blank">http://gigadb.org/site/guide</a>YesAdditional CommentsTo the best of my knowledge. Not my area of expertise.Is the data acquisition clear, complete and methodologically sound?NoAdditional CommentsI was lacking information about the transcriptomic data (it says in line 44 that RNA was extracted) that was used for the annotation? Was RNA only extracted from the muscle? Maybe the caveats that go along with that should be discussed. How was the data processed? How many reads etc. I think the manuscript lacks information about this unless I misunderstood where the data for the "transcript-based prediction" came from. Then this should be indicated more clearly.Is there sufficient detail in the methods and data-processing steps to allow reproduction?YesAdditional CommentsTo the best of my knowledge. I am not an expert on this.
Minor comments:
l 46: Which protocol?
Is there sufficient data validation and statistical analyses of data quality? Not my area of expertiseAdditional CommentsOne thing that could be probably additionally done is to provide dot plots with the 1-2 more closely related species with chromosome level assemblies (probably Tilapia or Medaka). Is the validation suitable for this type of data?YesAdditional CommentsAs far as I can judge the analysis is fine. Is there sufficient information for others to reuse this dataset or integrate it with other data?YesAdditional CommentsGenome and annotation are available, which is the most important for reuse and integration with other data sets. So as far as I can judge there is sufficient information for others.Any Additional Overall Comments to the AuthorFrom my viewpoint, this is a useful chromosome-level genome, so I support its publication. Beyond being a useful resource, I was however a bit disappointed by the 'scientific part' regarding the bi-color body formation. While the pigmentation of the bicolor angelfish is certainly a very exciting phenotype, the analysis performed is far too superficial to give any solid insights into the phenotype. I would suggest the authors toning this down in title, abstract and main text. It is fine to mention this as a future research direction and to state that the performed initial analysis (fig. 6 and 7) might aid these investigations, but the data does not permit further conclusions. Especially as GigaByte does not focus on analyses for biological findings, this should be completely sufficient.RecommendationMajor Revision